# Effects of Arbuscular Mycorrhizal Fungi on Yield, Biochemical Characteristics, and Elemental Composition of Garlic and Onion under Selenium Supply

**DOI:** 10.3390/plants9010084

**Published:** 2020-01-09

**Authors:** Nadezhda Golubkina, Zarema Amagova, Visita Matsadze, Svetlana Zamana, Alessio Tallarita, Gianluca Caruso

**Affiliations:** 1Federal Scientific Center of Vegetable Production, Selectsionnaya str. 14, VNIISSOK, Odintsovo District, Moscow 143072, Russia; 2Chechen Scientific Institute of Agriculture, Lenina 1, Grozny 366021, Grozny Region, Chechen Republic, Russia; amman1999@mail.ru (Z.A.); macaev58@mail.ru (V.M.); 3Department of Agriculture and Crop Production, State University of Land Management, Kazakova str. 15, Moscow 105064, Russia; svetlana.zamana@gmail.com; 4Department of Agricultural Sciences, University of Naples Federico II, 80055 Portici (Naples), Italy; lexvincentall@gmail.com (A.T.); gcaruso@unina.it; (G.C.)

**Keywords:** *Allium sativum* L., *Allium cepa* L., arbuscular mycorrhizal fungi, selenium, antioxidants, mineral content

## Abstract

Biofortification of garlic and onion plants with selenium and arbuscular mycorrhizal fungi inoculation are considered beneficial for producing functional food with anticarcinogenic properties. The effects of arbuscular mycorrhizal fungi (AMF) inoculation, sodium selenate foliar application, and the combination AMF + selenium (Se), compared to an untreated control, were assessed regarding the bulb yield, biochemical characteristics, and mineral composition. AMF + Se application resulted in the highest yield, monosaccharides, and Se content in both garlic and onion bulbs; and an increase of ascorbic acid and flavonoids in onion, and flavonoids in garlic. The highest bulb concentrations of P and K were recorded under the AMF + Se application, Ca was best affected by AMF, and Mg accumulation was highest under all the treatments in garlic and upon AMF + Se application in onion bulbs. Contrary to garlic, onion bulbs were characterized by a remarkable increase in microelements (B, Cu, Fe, Mn, Si, Zn) under the AMF + Se treatment. Selenium, either with or without AMF application, promoted the accumulation of B and Si in onion and Mo and Zn in garlic bulbs. It can be inferred that the interaction between AMF and Se is highly specific, differing for garlic and onion grown in similar environmental conditions in Grozny region, Chechen Republic.

## 1. Introduction

Among the *Allium* species, garlic and onion are the most widespread vegetables worldwide, and show a high nutritional value and beneficial effect on human health. These crops are extremely popular in the Chechen Republic, where the population suffer from intensive oxidative stress associated with the consequences of hostilities [1]. The socio-psychological instability of the population, and the serious environmental problems associated with environmental pollution by heavy metals, oil spills, soil degradation, and deforestation require urgent actions, including optimization of human nutrition. Garlic and onion possess high antioxidant activity, which may be significantly improved via fortification with selenium, thus increasing the efficiency of the antioxidant defense. In addition, *Allium sativum* and *Allium cepa* plants have the ability to convert the inorganic forms of selenium to well-known anticarcinogenic methylated forms of Se-containing amino acids [2], thus stimulating investigations relevant to the effects of selenium biofortification on *Allium* crops. Garlic and onion belong to secondary selenium accumulators, showing a remarkable tolerance to high concentrations of selenium due to the ability to synthesize Se-containing compounds, accordingly preventing the incorporation of this element into biologically active proteins; the latter phenomenon causes the decrease of enzymes’ biological activity as well as toxicity in other species [3].

Selenium is a powerful natural antioxidant participating in human organism protection against cardiovascular and viral diseases, and cancer, and improves immunity, fertility, and brain activity [4]. Nevertheless, Se is not essential to plants, though it is involved in the defense mechanisms against different forms of oxidative stress [3].

Despite the great prospects of producing Se-enriched vegetables in order to cure widespread Se deficiency [5], a serious ecological problem connected with the low efficiency of exogenous Se accumulation exists. Moreover, despite the sufficient Se concentrations in the soils of many areas from around the world, Se deficiency arises due to low soil Se bioavailability to plants. In this respect, inoculation of arbuscular mycorrhizal fungi (AMF) may be a promising tool for coping with this issue.

AMF are beneficial fungi forming a symbiotic association with many terrestrial plants, improving nutrient and water uptake thanks to an increased root surface through the wide hyphae system. The efficiency of AMF inoculation is governed by the type of AMF, plant species, soil nutrient bioavailability to plants, climate, and stress factors [6,7]. The limited root apparatus expansion in *Allium* plants, compared to most other genera, makes them extremely reactive to mycorrhizal fungi, thus allowing a decrease in the mineral fertilizer supply [8]. Notably, AMF enhance the uptake of Ca, Zn, S, Cu, Fe, and Mn, in addition to phosphorous, nitrogen, and potassium [9]. 

Controversial results have been published about the effects of AMF on selenium accumulation in plants. In this respect, Larsen et al. [10] reported an increase of Se content in garlic by supplying AMF solely or in combination with sodium selenate, differently from Patharajan and Raaman [11], who did not detect a positive effect on Se accumulation upon the plant’s joint supply with AMF and SeO_2_. The latter authors also reported the depression of AMF sporulation in soils enriched with selenium dioxide. Moreover, the application of the AMF *Glomus claroideum* in combination with bacteria tolerant to Se (*Stenotrophomona* sp. B19, *Enterobacter* sp. B16, *Bacillus* sp. R12., and *Pseudomonas* sp. R8) resulted in a 23% Se content increase in wheat [12]. In other investigations [13], the AMF inoculation to shallot plants concurrently with Se fortification stimulated the plant’s absorption of selenium from both organic and inorganic Se forms. The opposite results were recorded in lettuce [14], as well as in maize, alfalfa, and soybean [15], where the arbuscular mycorrhizal fungi reduced the plant’s Se content. 

The present research aimed to assess the effects of AMF inoculation in interaction with selenium fortification on the bulb yield, biochemical indicators and elemental composition of garlic and onion bulbs grown in Grozny region, Chechen Republic.

## 2. Results and Discussion

### 2.1. Root Mycorrhizal Colonization, Bulb Yield, and Dry Matter

In the present research carried out in Chechen Republic, the root mycorrhizal colonization did not significantly change between the two determinations performed one month after the transplant and at the end of the crop cycles and, therefore, the mean values of the two determinations are reported in Table 1. This parameter was significantly affected by the AMF inoculation, showing a 310% and 327% increase in the inoculated plant roots of garlic and onion, respectively, compared to the non-inoculated control. Differently, the selenium treatment did not have a significant effect on the root mycorrhizal colonization, and the combination between AMF and selenium showed a tendency to enhance the plant–microorganism symbiosis compared to sole AMF. The aforementioned findings are consistent with previous reports relevant to the application of AMF and selenium on shallot [13].

The increase of onion and garlic bulb yield due to AMF inoculation reached 1.45 to 1.50 times either with or without sodium selenate fortification (Table 2), suggesting the predominant role of AMF in bulb weight formation. A tendency of the bulb yield to increase upon joint application of AMF and Se was recorded both in garlic and onion. Spraying sodium selenate without concurrently inoculating AMF did not affect the garlic and onion mean bulb weight and, accordingly, the yield (Table 1). In previous research, mycorrhizal inoculation with *Glomus versiforme* and *Glomus intraradices* was found to improve onion growth by enhancing the leaf area index, shoot dry weight, leaf chlorophyll content, bulbing precocity [16], bulb yield, and water use efficiency [17], under both well-watered and water-deficit conditions [18]. In contrast with the results of the present investigation, Patharajan and Raaman [11] reported a decrease of the garlic bulb weight as a result of selenium treatment, and this divergence may reflect the diverse management of their trial: Se application to soil, a different Se form (SeO_2_), and higher Se doses. Indeed, Patharajan and Raaman [11] reported that Se treatment to soil did not promote the mycorrhizal association, and the mycorrhizal plants were not encouraged to uptake and accumulate Se under Se application.

As it can be observed in Table 1, the bulb dry matter was more affected by selenate application in garlic and by AMF inoculation in onion. These facts may reflect the higher tolerance to selenium supply showed by garlic compared to onion [3] and the less developed roots in onion, which are highly dependent on AMF inoculation. Other researchers have reported higher shoot and root dry weights of AMF plants than those of non-AMF plants at low or high temperature conditions [19,20], which is consistent with the results of the present research.

### 2.2. Carbohydrates and Organic Acids Content

The data reported in Table 2 suggest the similarity of the carbohydrate profile in garlic and onion bulbs elicited by AMF and Se treatments. Indeed, AMF inoculation increased the total carbohydrate content by 20% in garlic and by 11% in onion compared to the control, whereas the total sugar content was enhanced by 21.8% and 6% as a result of joint application of sodium selenate and AMF in *A. sativum* and *A. cepa*, respectively. Se treatment stimulated the accumulation of total sugars only in garlic, which is in accordance with the higher tolerance of garlic to high Se levels compared to onion [3].

Changes in the monosaccharides content were much more significant compared to total sugars. A synergic effect of AMF and Se was recorded in garlic: The increase in the monosaccharides concentration reached 75% as a result of joint application of Se and AMF, whereas the sole AMF inoculation increased these compound levels only by 8%. These results are in agreement with the positive effect of both Se and AMF on photosynthesis. In this respect, AMF reportedly modulate photosynthesis and the sugar concentration [11,21], in particular via phytohormone induction (for instance, abscisic acid) [22,23], as it has been described in different plants [6,24,25,26]. As far as selenium is concerned, this element may have a positive effect on yield formation via participation in chlorophyll biosynthesis and carbohydrate metabolism enhancement, and to some extent via the antioxidative effects of Se [3].

The applied treatments showed an even higher effect on onion bulbs: The AMF + Se application resulted in a 200% increase of the monosaccharides content while the sole AMF inoculation resulted in a 68% increase. In both garlic and onion, the foliar application of sodium selenate led to a less pronounced effect compared to the joint AMF + Se treatment (29% vs. 41%, respectively; Figure 1). In previous research [13], the joint application of AMF and sodium selenate on shallot plants also increased the bulb monosaccharides content by 1.8 times, referring to the total amount of sugars. However, contrary to *A. cepa*, shallot did not show any changes in the total sugar content as a result of the AMF and Se treatment. From the above reports, it can be inferred that the increase in the monosaccharides content due to joint application of AMF and sodium selenate occurs in at least three *Allium* species: garlic, onion, and shallot. The higher content of monosaccharides in AMF-inoculated plants compared to controls may be explained by the physiological changes induced by AMF, particularly the enzyme activity increase, which converts the complex sugars into simple ones. 

The biosynthesis of organic acids is closely connected with sugar metabolism. Regvar et al. [27] reported an increase in the total acidity of tomato fruits as a result of AMF application. The same phenomenon was also described in onion by Rospadek et al. [28], who also found that the total acidity increase under AMF application was accompanied by significant changes in the content of organic acid rankings as follows: from malic > propionic > tartaric > valeric > citric to malic > propionic > citric > valeric > tartaric.

In the present research, compared to onion, garlic was characterized by significantly lower levels of organic acids, and the most pronounced increase in total acidity was recorded only upon the joint application of AMF and Se. On the contrary, in onion, the highest increase in the TA value corresponded to separate AMF and Se applications while the AMF + Se supply to plants did not result in statistically significant differences compared to the untreated control. A lack of literature reports on AMF–Se interaction on this quality indicator makes it difficult to explain the mentioned phenomenon.

### 2.3. Antioxidants

The results stemming from the present work are in accordance with the reports from several investigations regarding the beneficial effect of AMF inoculation on the antioxidant status of plants, including onion and leek [8,28,29,30]. In similar growing conditions to those arranged in our study, the antioxidant status showed peculiar changes as a function of the species examined, i.e., garlic or onion (Table 3 and Table 4): The AMF inoculation had a higher beneficial effect on onion than on garlic.

Notably, the AMF inoculation of onion resulted in an increase of the bulb phenolics under all treatments (AMF, Se, AMF + Se) compared to the control, whereas no significant changes were recorded in garlic. The highest beneficial effect on ascorbic acid, flavonoids, phenolics concentrations, and total antioxidant activity was elicited by the joint application of AMF + Se, whereas the latter treatment led to a slight increase of the flavonoids content in garlic. Notably, Se biofortification without AMF application resulted in the stimulation of ascorbic acid and polyphenols biosynthesis in onion, but no effect was detected in garlic.

Interestingly, onion plants were more reactive to AMF and/or Se application in increasing antioxidant activity, organic acids, and dry matter content compared to garlic, but both species showed a pronounced increase in the monosaccharides content under the mentioned treatments (Table 4).

### 2.4. Selenium Accumulation

Being a chemical analog of sulfur, selenium freely substitutes S in biological systems. Earlier investigations demonstrated that garlic bulbs contain higher levels of sulfur than onion: About 0.46 to 0.60% d.w. vs. 0.154 to 0.535% d.w. [31]. This phenomenon is in agreement with the fact that garlic can accumulate up to five times more selenium than onion (110–150 mg·kg^−1^ vs. 28 mg·kg^−1^), which makes *A. sativum* a more powerful natural anticarcinogen [32]. The present results confirm the aforementioned statement, i.e., significantly higher concentrations of the element in garlic bulbs than in onion (Figure 2).

The data are also in accordance with the beneficial effect of AMF on Se accumulation in ordinary conditions and under an Se supply described in garlic [10] and shallot [13]. Our results show that the consumption of 5 g of Se-fortified garlic bulbs obtained upon application of sodium selenate and AMF may provide about 50% of the adequate Se consumption level (ACL), which is equal to 70 µg per day. Higher levels of onion consumption compared to garlic suggest that 50 g of Se-enriched onion will be equal to 1.4 ACL of Se.

In the absence of exogenous Se, AMF inoculation provides a 5-fold increase of the Se content in garlic, 10-fold in onion, and, as previously mentioned, 8-fold in shallot bulbs [13]. Notably, contrary to garlic and onion, shallot bulbs accumulate significantly lower levels of selenium when treated with Se and AMF, with values not exceeding 5000 µg·kg^−1^ d.w. Interestingly, the beneficial effect of AMF application on Se accumulation in *Allium* plants widely depends on the species, being higher in garlic compared to *A. cepa* and shallot. 

### 2.5. Macroelements

In the present research, AMF inoculation resulted in a reduction of the bulb nitrate concentration both in garlic and onion; however, the latter effect was not recorded under the joint application AMF + Se. Other authors reported an improved plant assimilation of nitrogen, which enhances plant growth and development upon AMF inoculation [6,33] as well as the yield and quality of edible parts in *Allium* species [34,35]. Kucova et al. [35] reported that AMF decrease nitrate leaching along the soil profile. The reduction of the nitrate concentration upon AMF inoculation in garlic and onion reflects the process of nitrate-reductase activation by these fungi [6,36].

AMF and Se application led to different effects on the mineral content in garlic and onion bulbs. In fact, the K and P concentration increase caused by AMF inoculation was more pronounced in onion than in garlic; the opposite trend was detected for Mg, which showed an augmentation of almost 200% in garlic bulbs under AMF, Se, and AMF + Se treatments, and only a 22% rise in onion upon the joint application AMF + Se (Table 5, Figure 3). In the conditions of the Chechen Republic, AMF inoculation also resulted in a Ca content increase both in garlic and onion. Conversely, the application of Se, solely or in combination with AMF, did not affect Ca accumulation.

The higher tolerance of garlic plants to salt stress compared to onion [37] and the known Se ability to increase Na content in plants [38] may be connected with the observed differences in sodium level changes caused by AMF and Se applications (Figure 3).

### 2.6. Microelements

The most significant differences in garlic and onion reactivity to AMF and Se application referred to the accumulation of trace elements in the bulbs. The data reported in Table 6 demonstrate that the most remarkable increase in B, Cu, Fe, Mn, Mo, Si, and Zn content was recorded in onion bulbs under the joint application of AMF + Se. Differently, among the aforementioned elements, only Mo and Zn concentrations were higher in garlic bulbs supplied with AMF + Se or Se compared to the untreated control. The latter finding entails that selenium’s effect on the mineral composition is more pronounced in garlic than in onion, the latter being more reactive to AMF inoculation.

The beneficial effect of Se fortification, either combined or not with AMF, was recorded on the increase of the B and Si concentration in onion and of Mo and Zn in garlic. Previous reports relevant to AMF inoculation suggest the chance to increase the boron content [39], though this does not explain the differences in bulb B accumulation between garlic and onion.

Unexpectedly, wide differences in bulb Mo concentrations were recorded, as all treatments increased the Mo content in garlic, whereas only the joint application of AMF + Se showed a beneficial effect on onion. Furthermore, AMF inoculation even decreased the Mo level in onion bulbs (Figure 4, Table 6). The beneficial effect of AMF on Mo accumulation was also described in a previous work on maize [40]. Notably, molybdenum is an essential component in the active site of several enzymes: nitrate reductase, which reduces nitrate to nitrite; xanthine dehydrogenase, which is required for purine degradation; sulfite oxidase, which catalyzes the oxidation of sulfite (SO_3_^−2^) to sulfate (SO_4_^−2^); and aldehyde oxidase, which is required for the synthesis of abscisic acid (ABA). To date, no information regarding the effect of AMF on Mo accumulation in *Allium* species plants has been published. 

Interestingly, an extremely high increase of the Fe concentration in onion bulbs was detected upon the AMF + Se application and significantly lower levels of this element in *Allium cepa* inoculated with AMF. Conversely, no beneficial effect was recorded in garlic.

Contrary to garlic, onion was characterized by a significant increase of the Si and Cu levels as a result of AMF inoculation either jointly or separately from the Se supply (Table 7).

## 3. Material and Methods

### 3.1. Plant Material and Experimental Conditions

Research was carried out on a garlic (*Allium sativum* L.) cultivar Maysky and onion (*Allium cepa* L.) cultivar Kaba at the experimental fields of the Chechen Scientific Institute of Agriculture, Grozny region, Chechen Republic (43°19′ N, 45°42′ E) in 2017–2018 and 2018–2019. The trial was conducted on a leached chernozems soil of the forest-steppe zone of the Chechen Republic, characterized by pH 6.9, 3%–5% humus content, 108 mg·kg^−1^ N, 132 mg·kg^−1^ P_2_O_5_, and 145 mg·kg^−1^ K_2_O. The mean values of the temperature (°C) and relative humidity (%) as an average of 2017–2018 and 2018–2019 were as follows: 0.7 and 87.3 in November; 1.3 and 90.6 in December; 0.2 and 91.5 in January; 1.5 and 83.7 in February; 5.6 and 75.7 in March; 17.2 and 61.5 in April; 18 and 74.5 in May; 24 and 62.9 in June; 23.4 and 67.7 in July; and 24.7 and 61.0 in August.

The experimental protocol was based on a comparison between four treatments: (1) Soil inoculation of AMF-based formulate (Rhizotech Plus at 2 g·m^−2^ soil); (2) foliar supply of sodium selenate (150 mg·m^−2^, 63 mg Se m^−2^ by 50 mg·L^−1^ 0.26 mM solution); (3) combined application of AMF and sodium selenate (Na_2_SeO_4_); and (4) control, with neither AMF nor Se application. 

A randomized complete blocks design with three replicates was used for the treatments’ distribution in the field, with each treatment covering a 12-m^2^ (4 × 3 m) surface area. 

Garlic was planted on 23 November both in 2017 and 2018, with an 8-cm spacing along rows, which were 20 cm apart. The AMF were applied twice, just before planting and on 30 April (the beginning of bulb formation). Se foliar fortification was performed 3 times, at 10-day intervals starting from 7 May. Garlic plants were harvested on 18 July. 

Onion was planted on 27 April with a 10-cm spacing along rows, which were 20 cm apart. The AMF inoculation was performed twice, just before planting and on 23 May (the beginning of bulb formation). Sodium selenate was sprayed three times at 10-day intervals starting from 27 May. Onion plants were harvested on 2 August. 

Fertilization of both species crops was performed according to Caruso et al.’s [41] recommendations: At planting, with 42 kg·ha^−1^ N as ammonium sulphate, 48 kg·ha^−1^ P_2_O_5_ as superphosphate, and 78 kg·ha^−1^ K_2_O as potassium sulphate; at the beginning of bulb formation, with 78 kg·ha^−1^ N and 84 kg·ha^−1^ K_2_O as calcium nitrate and potassium nitrate. Drip irrigation was activated when the soil’s available water capacity decreased below 70%. Treatments with copper oxychloride were practiced against rust. 

Root mycorrhizal colonization (as a percentage) was assessed twice, one month after planting and at the crop cycle’s end, according to the Giovannetti and Mosse method [42].

### 3.2. Sample Preparation

After harvesting, the sampled bulbs were cleaned, also removing the broken outer shells, and cut with a plastic knife to form thin slices. Fresh homogenized material was used for nitrates and ascorbic acid determination. An aliquot of fresh slices was dried at 70 °C to a constant weight and used for assessing the antioxidant activity, polyphenols, flavonoids, and elemental composition. The selenium content was determined in homogenized samples dried at 20 °C up to a constant weight to minimize Se losses. 

### 3.3. Antioxidants

#### 3.3.1. Ascorbic Acid

It was determined by visual titration of plant extracts in 6% trichloracetic acid with Tillmans reagent [43]. Three grams of fresh garlic clove or onion bulb homogenates were ground in porcelain mortar with 5 mL of 6% trichloracetic acid and quantitatively transferred to a measuring cylinder. The volume was brought to 60 mL using trichloracetic acid, and the mixture was filtered through filter paper 15 min later. The concentration of ascorbic acid was determined from the amount of Tillmans reagent that went into the titration of the sample. 

#### 3.3.2. Polyphenols

Polyphenols were determined in water (garlic) and 70% ethanol (onion) extracts, using the Folin–Ciocalteu colorimetric method as described by Golubkina et al. [44] with minor modifications. One gram of dry garlic powder was extracted with 20 mL of distilled water at room temperature for 2 h. Dry onion bulb homogenates were extracted with 70% EtOH at 80 °C for 1 h. The mixtures were quantitatively transferred to a volumetric flask and the volume was adjusted to 25 mL. The mixture was filtered through filter paper and 1 mL of the resulting solution was transferred to a 25-mL volumetric flask, to which 2.5 mL of saturated Na_2_CO_3_ solution and 0.25 mL of diluted (1:1) Folin–Ciocalteu reagent were added, and the volume was brought to 25 mL with distilled water. One hour later, the solutions were analyzed through a spectrophotometer (Unico 2804 UV, Dayton, NJ, USA) and the concentration of polyphenols was calculated according to the absorption of the reaction mixture at 730 nm. As an external standard, 0.02% gallic acid was used.

#### 3.3.3. Flavonoids

The total flavonoids content was determined by the spectrophotometric method, based on the flavonoid-aluminum chloride (AlCl_3_) complexation [45]. The standard of quercetin dehydrate (98% HPLC) was purchased from Sigma Co. (St. Louis, MO, USA). In total, 10 mL of methanol were added to 1 g of dried and homogenized samples and the latter were left at room temperature for 2 h. The resulting mixture was filtered through a pleated filter. Then, 0.2 mL of the extract were diluted with 1.8 mL of methanol, and then 0.1 mL of 2% AlCl_3_, 0.5 mL of 1 M sodium acetate solution, and 1 mL of distilled water were added. The mixture was incubated for 30 min at room temperature, and the absorption at 415 nm was measured. The total flavonoid content was determined by a standard curve built using five different concentrations of quercetin–AlCl_3_ complex, using quercetin purchased from Fluka (Buchs, Switzerland).

#### 3.3.4. Antioxidant Activity (AOA)

The antioxidant activity of garlic cloves or onion bulbs was assessed using the redox titration method [46], via titration of 0.01 N KMnO_4_ solution with water/ethanol plant extracts. The reduction of KMnO_4_ to colorless Mn^+2^ in this process reflects the amount of antioxidants dissolvable in water/70% ethanol. The values were expressed in mg GAE g^−1^ d.w. The use of KMnO_4_ acidic solution is known to be successfully used for the determination of the *Ocimum basilicum* antioxidant potential [47,48] and antioxidant capacity of serum [49].

### 3.4. Nitrates

Nitrate content was assessed using an ion selective electrode by ionomer Expert-001 (Econix, Moscow, Russia). Five grams of fresh garlic clove or onion bulb homogenates were homogenized with 50 mL of distilled water. In total, 45 mL of the resulting extract were mixed with 5 mL of 0.5 M potassium sulfate background solution (necessary for regulating the ionic strength) and analyzed through the ionomer for nitrate determination.

### 3.5. Sugars

Monosaccharides were determined using the ferricyanide colorimetric method, based on the reaction of monosaccharides with potassium ferricyanide [50]. Total sugars were analogically determined after acidic hydrolysis of water extracts with 20% hydrochloric acid. Fructose was used as an external standard.

### 3.6. Selenium

Selenium was analyzed using the fluorometric method previously described for tissues and biological fluids [51]. Dried homogenized samples were digested via heating with a mixture of nitric-chloral acids, subsequent reduction of selenate (Se^+6^) to selenite (Se^+4^) with a solution of 6 N HCl, and the formation of a complex between Se^+4^ and 2,3-diaminonaphtalene. Calculation of the Se concentration was achieved by recording the piazoselenol fluorescence value in hexane at 519 nm λ emission and 376 nm λ excitation. Each determination was performed in triplicate. The precision of the results was verified using a reference standard-lyophilized cabbage in each determination with a Se concentration of 150 μg·Kg^−1^.

### 3.7. Elemental Composition

B, Ca, Cu, Fe, K, Mg, Mn, Mo, Na, P, Si, and Zn contents in garlic and onion bulb samples were assessed using ICP-MS on a quadruple mass-spectrometer Nexion 300D (Perkin Elmer Inc., Shelton, CT, USA), equipped with the seven-port FAST valve and ESI SC DX4 autosampler (Elemental Scientific Inc., Omaha, NE, USA) at the Biotic Medicine Center (Moscow, Russia). Rhodium 103 Rh was used as an internal standard to eliminate instability during measurements. Quantitation was performed using an external standard (Merck IV, multi-element standard solution), potassium iodide for iodine calibration, Perkin–Elmer standard solutions for P and Si, and all the standard curves were obtained at five different concentrations. For quality control purposes, the internal controls and reference materials were tested together with the samples on a daily basis. Microwave digestion of samples was performed according to the standard method [52] with sub-boiled HNO_3_ (Fluka #02650 Sigma-Aldrich, Co.) in a Berghof SW-4 DAP-40 microwave system (Berghof Products + Instruments GmbH, Eningen, Germany), diluted 1:150 with distilled deionized water. The instrument conditions and acquisition parameters were as follows: plasma power and argon flow, 1500 and 18 L min^−1^ respectively; aux argon flow, 1.6 L min^−1^; nebulizer argon flow, 0.98 L min^−1^; sample introduction system, ESI ST PFA concentric nebulizer and ESI PFA cyclonic spray chamber (Elemental Scientific Inc., Omaha, NE, USA); sampler and slimmer cone material, platinum; injector, ESI Quartz 2.0 mm I.D.; sample flow, 637 µL min^−1^ ; internal standard flow, 84 µL min^−1^; dwell time and acquisition mode, 10–100 ms and peak hopping for all analytes; sweeps per reading, 1; reading per replicate, 10; replicate number, 3; DRC mode, 0.55 mL min^−1^ ammonia (294993-Aldrich Sigma-Aldrich, Co., St. Louis, MO, USA) for Ca, K, Na, and Fe optimized individually for RPa and RPq; and STD mode, for the rest of analytes at RPa = 0 and RPq = 0.25.

### 3.8. Statistical Analysis

Data were processed by analysis of variance and mean separations were performed through the Duncan multiple range test, with reference to the 0.05 probability level, using SPSS software version 21. Data expressed as a percentage were subjected to angular transformation before processing. 

## 4. Conclusions

The results obtained in the present research show that AMF inoculation led to an enhanced bulb yield from both garlic and onion crops. Moreover, the effects of either separate or joint application of AMF and selenium on the biochemical characteristics and mineral content of garlic and onion bulbs grown in similar environmental conditions were very different depending on the *Allium* species. A positive stimulating effect of AMF on Se accumulation was recorded in both garlic and onion bulbs, though the overall patterns relevant to mineral element changes upon the application of the experimental treatments were rather complex.

As the effects of selenium fortification and AMF treatment are deeply influenced by genetic traits and growth conditions, the comparison performed in the present research between garlic and onion behavior under AMF and/or selenate treatments, in similar ecological conditions, can be considered as an important example of responses from *Allium sativum* and *Allium cepa* plants, which are known to be secondary selenium accumulators. Indeed, such responses depend on: the different plants’ ability to accumulate macroelements and trace elements; variations in the activity of relevant enzymes participating in antioxidant defense and carbohydrates or organic acids synthesis; differences in the tolerance to high concentrations of selenium; and variations in the root structure. 

The outcomes of this investigation highlight the effectiveness of two sustainable farming strategies, such as AMF inoculation and plant selenium biofortification, in enhancing garlic and onion performances: the arbuscular mycorrhizal fungi increased the bulb yield, and both AMF and Se improved the overall quality, antioxidant, and mineral element indicators of the two *Allium* species examined, compared to the untreated control.

## Figures and Tables

**Figure 1 plants-09-00084-f001:**
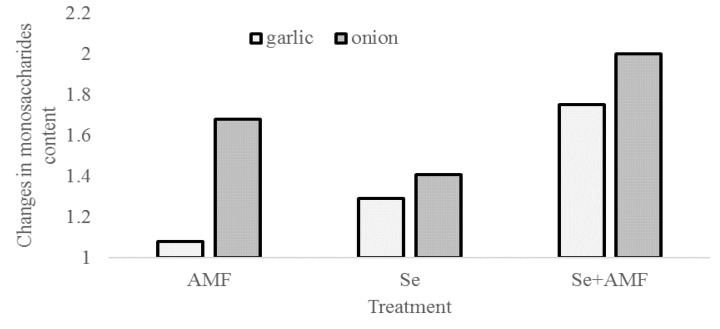
Changes in monosaccharides content in garlic and onion bulbs, compared to control, as affected by AMF and sodium selenate.

**Figure 2 plants-09-00084-f002:**
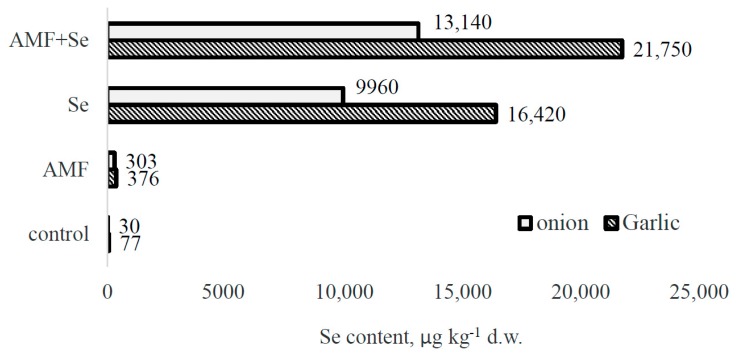
Effect of AMF on Se accumulation in garlic and onion bulbs.

**Figure 3 plants-09-00084-f003:**
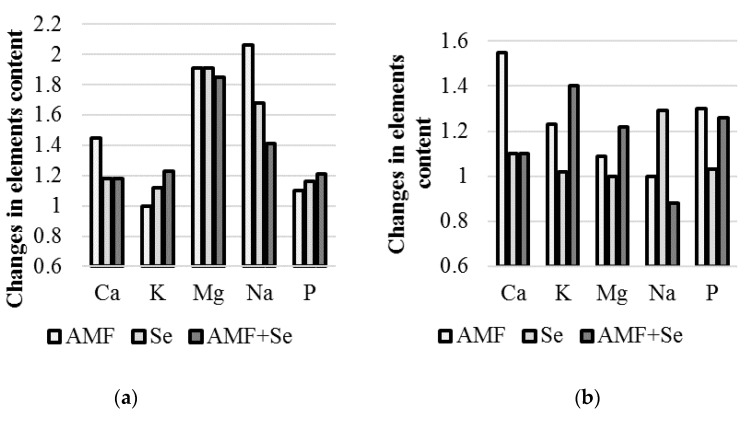
Changes in the macroelement concentration, compared to the untreated control, as affected by AMF and Se applications: (**a**) garlic, (**b**) onion.

**Figure 4 plants-09-00084-f004:**
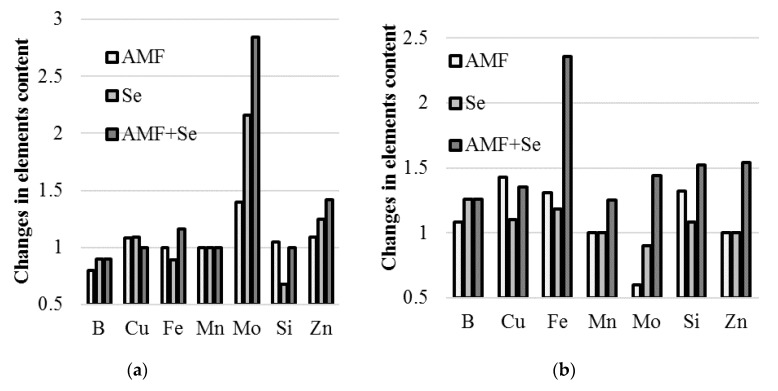
Changes in the microelement content, compared to the untreated control, in garlic (**a**) and onion (**b**) bulbs as affected by AMF and Se application.

**Table 1 plants-09-00084-t001:** Effect of arbuscular mycorrhizal fungi (AMF) and selenium application on root mycorrhizal colonization, yield, mean bulb weight, and dry matter in garlic and onion.

Species	Treatment	Root Mycorrhizal Colonization (%)	Yield (t·ha^−1^)	Mean Bulb Weight (g)	Dry Matter (%)
Garlic	Control	21.3 ± 2.2 b	11.5 ± 1.3 c	18.4 ± 2.1 c	34.8 ± 0.1 c
	AMF	65.7 ± 3.0 a	16.1 ± 1.9 b	25.8 ± 4.3 b	37.6 ± 0.2 a
	Se	21.5 ± 2.0 b	11.8 ± 1.8 c	18.9 ± 3.7 c	38.0 ± 0.1 a
	AMF + Se	66.4 ± 2.8 a	17.9 ± 2.2 a	28.7 ± 4.7 a	36.1 ± 0.3 b
Onion	Control	21.1 ± 2.3 b	22.7 ± 3.0 b	45.3 ± 7.8 b	14.3 ± 0.2 b
	AMF	68.6 ± 2.7 a	33.0 ± 3.5 a	65.9 ± 7.2 a	17.0 ± 0.4 a
	Se	21.7 ± 2.2 b	22.8 ± 2.9 b	45.5 ± 7.6 b	15.0 ± 0.4 b
	AMF + Se	69.2 ± 3.1 a	34.1 ± 3.5 a	68.1 ± 7.0 a	16.3 ± 0.1 a

Within each column and species, values followed by different letters are significantly different according to Duncan test at *p* ≤ 0.05.

**Table 2 plants-09-00084-t002:** Effect of AMF and selenium application on carbohydrates, total soluble solids, and titratable acidity in garlic and onion bulbs.

Species	Treatment	Monosaccharides (% d.w.)	Total Sugars (% d.w.)	Titratable Acidity (mg-eq Citric Acid 100 g^−1^ d.w.)
Garlic	Control	15.7 ± 1.1 c	50.9 ± 3.1 c	4.9 ± 0.2 b
	AMF	17.0 ± 1.2 c	60.9 ± 3.6 ab	4.3 ± 0.2 c
	Se	20.3 ± 1.3 b	59.8 ± 3.2 b	4.5 ± 0.2 bc
	AMF + Se	27.5 ± 1.6 a	62.0 ± 3.7 a	5.8 ± 0.3 a
Onion	Control	3.4 ± 0.3 c	61.8 ± 2.8 c	13.4 ± 1.0 b
	AMF	5.7 ± 0.5 ab	68.9 ± 3.1 a	19.4 ± 1.2 a
	Se	4.8 ± 0.4 b	63.7 ± 3.5 bc	17.6 ± 1.2 a
	AMF + Se	6.8 ± 0.5 a	65.2 ± 3.7 b	14.9 ± 1.1 b

Within each column and species, values followed by different letters are significantly different according to Duncan test at *p* ≤ 0.05.

**Table 3 plants-09-00084-t003:** Antioxidant compounds and activity in garlic and onion bulbs as affected by AMF and selenium application.

Species	Treatment	AOA (mg-eq GA g^−1^ d.w.)	Phenolics (mg-eq GA g^−1^ d.w.)	Flavonoids (mg-eq Q 100 g^−1^ d.w.)	Ascorbic Acid (mg 100 g^−1^ d.w.)
Garlic	Control	39.0 ± 1.1 b	5.3 ± 0.4	1.1 ± 0.1	59.8 ± 1.7 b
	AMF	38.8 ± 1.0 b	4.4 ± 0.4	1.1 ± 0.1	61.4 ± 1.8 b
	Se	40.5 ± 1.1 ab	5.2 ± 0.4	1.2 ± 0.1	62.9 ± 1.8 ab
	AMF + Se	42.2 ± 1.1 a	5.0 ± 0.3	1.3 ± 0.1	66.0 ± 2.1 a
			n.s.	n.s.	
Onion	Control	31.7 ± 0.7 c	11.5 ± 0.5 b	3.2 ± 0.1 b	59.4 ± 1.4 c
	AMF	36.4 ± 0.9 b	15.2 ± 0.6 a	3.4 ± 0.1 b	58.0 ± 1.3 c
	Se	33.0 ± 0.7 c	14.9 ± 0.6 a	3.2 ± 0.1 b	66.7 ± 1.5 b
	AMF + Se	43.4 ± 0.6 a	13.0 ± 0.5 b	5.4 ± 0.1 a	73.6 ± 1.5 a

n.s., not significant. Within each column and species, values followed by different letters are significantly different according to Duncan test at *p* ≤ 0.05. AOA, antioxidant activity, GA, gallic acid; Q, quercetin.

**Table 4 plants-09-00084-t004:** Significant effects of AMF and Se on yield, dry matter, sugar, and antioxidants content of onion and garlic bulbs, compared to the untreated control.

Treatment	Bulb Weight	Dry Matter	Monosaccharides	Total Sugars	TA	AA	Fl	PP	AOA
**Garlic**									
AMF	*			*					
Se			*	*					
AMF + Se	**		*	*			*		
**Onion**									
AMF	*	*	*		*			*	
Se			*		*	*		*	
AMF + Se	**	*	*			**	**	*	*

* indicates the significant difference at *p* ≤ 0.05, compared to the untreated control. ** indicates the significant difference at *p* ≤ 0.001, compared to the untreated control; TA, titratable acidity; AA, ascorbic acid; Fl, flavonoids; PP, phenolics; AOA, antioxidant activity.

**Table 5 plants-09-00084-t005:** Macroelements content in garlic and onion as affected by AMF and Se.

Species	Treatment	Nitrates (mg·kg^−1^ d.w.)	Ca	K	Mg	Na	P
			(g·kg^−1^ d.w.)				
Garlic	Control	1483 ± 58 a	0.51 b	15.5 c	0.90 b	403 c	4.22 b
	AMF	1172 ± 37 b	0.74 a	14.2 c	1.72 a	832 a	4.63 ab
	Se	1404 ± 54 a	0.60 b	17.4 b	1.72 a	679 b	4.92 ab
	AMF + Se	1479 ± 55 a	0.60 b	19.1 a	1.67 a	567 b	5.13 a
Onion	Control	1310 ± 30 a	1.92 b	16.7 c	1.97 b	809 b	3.93 b
	AMF	1144 ± 39 b	2.97 a	20.6 b	2.15 b	748 b	5.10 a
	Se	1293 ± 42 a	2.13 b	17.0 c	1.93 b	1046 a	4.05 b
	AMF + Se	1261 ± 40 a	2.08 b	23.4 a	2.43 a	708 b	4.97 a

Within each column and species, values followed by different letters are significantly different according to Duncan test at *p* ≤ 0.05.

**Table 6 plants-09-00084-t006:** Microelement content in garlic and onion bulbs as affected by AMF and selenium.

Species	Treatment	B	Cu	Fe	Mn	Mo	Si	Zn
		(mg·kg^−1^ d.w.)						
Garlic	Control	8.9 a	6.4	58.2 ab	11.8	0.25 d	19.8 a	38.2 c
	AMF	7.0 b	6.9	57.9 ab	12.4	0.35 c	13.5 b	41.7 bc
	Se	8.0 ab	7.0	51.7 b	11.8	0.54 b	20.8 a	47.9 ab
	AMF + Se	8.0 ab	6.6	67.7 a	10.4	0.71 a	18.4 a	54.1 a
			n.s.		n.s.			
Onion	Control	8.7 b	6.3 b	40.7 c	9.1 ab	0.25 b	16.5 b	28.5 b
	AMF	9.4 b	9.0 a	53.5 b	8.9 b	0.15 c	17.9 b	30.0 b
	Se	11.0 a	6.9 b	48.1 c	8.8 b	0.22 b	21.7 a	27.8 b
	AMF + Se	11.0 a	8.5 a	96.1 a	11.4 a	0.36 a	25.0 a	43.9 a

n.s., not significant. Within each column and species, values followed by different letters are significantly different according to Duncan test at *p* ≤ 0.05.

**Table 7 plants-09-00084-t007:** Significant effects of AMF and Se on the mineral composition of garlic and onion bulbs, compared to the untreated control.

Treatment	B	Cu	Fe	Mn	Mo	Si	Zn	Ca	K	Mg	Na	P
**Garlic**												
AMF								*		*	*	
Se					*		*			*	*	
AMF + Se					*		*			*	*	*
**Onion**												
AMF		*	*					*	*			*
Se	*					*					*	
AMF + Se	*	*	*	*	*	*	*	*		*		*

* indicates the significant difference at *p* ≤ 0.05, compared to the untreated control.

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
