# Peer review of "Effects of Arbuscular Mycorrhizal Fungi on Yield, Biochemical Characteristics, and Elemental Composition of Garlic and Onion under Selenium Supply"

_plants, 2020, doi:10.3390/plants9010084_

Round 1

Reviewer 1 Report

The manuscript "Effects of Arbuscular Mycorrhizal fungi on yield biochemical characteristics and elemental composition of garlic and onion under selenium supply" by Golubkina et. al. investigates the effects of Arbuscular mycorrhizal fungi (AMF) inoculation in interaction with selenium on yields, carbohydrates, antioxidants and mineral content of garlic and onion bulbs. In this study, authors found AMF combined with Se could induce changes in various characteristics of garlic and onion. The studies of the administration on the Allium species are interesting. Nevertheless, the reviewer has the following concerns that need to be addressed before the acceptance of the manuscript.

Major comments: 
1. line269-272, author described the glucose and fructose may be increased in AMF inoculated plants; however, no evidence show the content of glucose and fructose in the current study. Therefore, identify the content of glucose and fructose may be conducted in this study.

2. Please describe/discuss why the levels of organic acids did not increase in the onion.

3.Authors need to briefly discuss the different outcomes in both garlic and onion after the same treatment.

Minor comments: 
1. Please recheck the author name of Ref 17 in the main text.

2. Please provide the detail information regarding each table.

    For instance, what do it mean about a, b and c in the table?

    In the Table 5 and Table 8, please provide the significant difference values instead of “+” or describe “+” presents the statistical difference range.

Author Response

Dear Reviewer 1, 

Thank you very much for your useful recommendations aimed to improve our manuscript. We addressed all your comments and performed the relevant amendments and/or modifications inside the text and tables, highlighting them in red color.

Answers to Reviewer 1 comments

Major comments:

line269-272, author described the glucose and fructose may be increased in AMF inoculated plants; however, no evidence show the content of glucose and fructose in the current study. Therefore, identify the content of glucose and fructose may be conducted in this study

Answer:

We are sorry for the inaccuracy in the description. Indeed, we did not analyze the carbohydrate profile in the present investigation, but we just referred to the literature reports about glucose and fructose. In order to make the description more convenient, we have changed the expression “The  higher  content  of  glucose  and  fructose … ” to “The  higher  content  of  monosaccharides …”. 

Please describe/discuss why the levels of organic acids did not increase in the onion.

Answer:

The discussion has been reported in the lines 295-300.

Authors need to briefly discuss the different outcomes in both garlic and onion after the same treatment.

Answer:

We have added the discussion requested in the Conclusion section, lines 418-437.

Minor comments: 

1. Please recheck the author name of Ref 17 in the main text.

Answer:

We are sorry for the misprint and we have replaced “Da Silva et al. [17]” with “Golubkina et al. [17]”.

Please provide the detail information regarding each table.

    For instance, what do it mean about a, b and c in the table?

In the Table 5 and Table 8, please provide the significant difference values instead of “+” or describe “+” presents the statistical difference range.

Answer:

We have highlighted in red colour the sentence describing the mean separation test “Duncan’s”, which is commonly used within the scientific community and entails matching a letter with a value based on the ranking relevant to the statistically significant differences at the probability level of 0.05 in our case.  

Tables 5 and 8 have been corrected according to the above recommendations.

Reviewer 2 Report

The present paper reports the results of a research aimed to evaluate the effects of AMF inoculation in combination to selenium supply on yield, biochemical traits and mineral compounds of garlic and onion bulbs. Authors found that the application of AMF and Se highly increases, yield, monosaccharides bulb content and Se accumulation in garlic and onion plants. For both vegetable crops the increase in antioxidant content was recorded with AMF+Se combination. In onion bulbs the highest levels of macro-elements (Са, Mg, P) were observed in AMF+Se combination. The same treatment positively affected also the content of macro-elements of bulbs much more for onion than for garlic. Authors concluded that the interaction between AMF and Se is highly specific in similar environmental conditions.

On overall, the subject falls within the general scope of the Journal. The paper is well written, the different sections of the manuscript are well established and well articulated, the title is appropriate, the abstract is quite informative. The results and discussion are clearly presented and well organized, the tables and figures are complete and self-explanatory, the conclusions and references are adequate.

Nevertheless, the manuscript needs minor changes and corrections as outlined below.

Abstract

Concise information about the experimental protocol should be reported.

The conclusion sentence should be completed with further short details.

Keywords: put in full and in italics style A. sativum” and “A. cepa” (Allium sativum, Allium cepa”.

Introduction

- Some information about the importance of Allium cepa L. and Allium sativum L. in the area where the research was carried out should be reported.

- A reference to the investigation area should be added to the research targets.

Materials and Methods

- Table 1 should be removed, and the temperature and relative humidity data should be included inside the text. 

Results and Discussion

- In Tables 5 and 8 the line sequence should be the same as in the other Tables: Control, AMF, Se, AMF+Se.

- Please, check if the combination between AMF and Se has been expressed uniformely as AMF+Se inside the text as well as the Tables and Figures captions.

References

- Some other references could be added and cited.  

Colella T., Candido V., Campanelli G., Camele I., Battaglia D., 2014. Effect of irrigation regimes and artificial mycorrhization on insect pest infestations and yield in tomato crop. Phytoparasitica, 42, 235-246. DOI: 10.1007/s12600-013-0356-3.            It could be cited in “Introduction” section” (line  59, along with citation n. 4).

Castronuovo D., Russo D., Libonati R., Faraone I., Candido V., Picuno P., Andrade P., Valentao P., Milella L., 2019. Influence of shading treatment on yield, morphological traits and phenolic profile of sweet basil (Ocimum basilicum L.). Scientia Horticulturae, 254, 91-98, doi: 10.1016/j.scienta.2019.04.077               It could be cited in “Material and Methods” section” (line  153, along with citation n. 20).

Author Response

Dear Reviewer 2, 

Thank you very much for your useful recommendations aimed to improve our manuscript. We addressed all your comments and performed the relevant amendments and/or modifications inside the text and tables, highlighting them in red color.

Answers to Reviewer 2 comments

The manuscript needs minor changes and corrections as outlined below.

Abstract

Concise information about the experimental protocol should be reported.

The conclusion sentence should be completed with further short details.

Answer:

The experimental details have been added and the conclusion has been expanded, according to the above recommendations.

Keywords: put in full and in italics style “ sativum” and “A. cepa” (Allium sativum, Allium cepa”.

Answer:

 We have performed the corrections in compliance with the above comments.

3) Introduction

- Some information about the importance of Allium cepa L. and Allium sativum L. in the area where the research was carried out should be reported.

- A reference to the investigation area should be added to the research targets.

Answer:

We have added (reference [1]) the information  relevant to the ecological frame in the Chechen republic and the importance of garlic and onion for the Chechen population diet.

Materials and Methods

- Table 1 should be removed, and the temperature and relative humidity data should be included inside the text. 

Answer:

We have deleted Table 1 and reported temperature and relative humidity values inside the text, as recommended above.

Results and Discussion

- In Tables 5 and 8 the line sequence should be the same as in the other Tables: Control, AMF, Se, AMF+Se.

- Please, check if the combination between AMF and Se has been expressed uniformly as AMF+Se inside the text as well as the Tables and Figures captions.

Answer:

We have performed corrections and check according to the above comments.

References

- Som

Dear Reviewer 2, 

Thank you very much for your useful recommendations aimed to improve our manuscript. We addressed all your comments and performed the relevant amendments and/or modifications inside the text and tables, highlighting them in red color.

Answers to Reviewer 2 comments

The manuscript needs minor changes and corrections as outlined below.

Abstract

Concise information about the experimental protocol should be reported.

The conclusion sentence should be completed with further short details.

Answer:

The experimental details have been added and the conclusion has been expanded, according to the above recommendations.

Keywords: put in full and in italics style “ sativum” and “A. cepa” (Allium sativum, Allium cepa”.

Answer:

 We have performed the corrections in compliance with the above comments.

3) Introduction

- Some information about the importance of Allium cepa L. and Allium sativum L. in the area where the research was carried out should be reported.

- A reference to the investigation area should be added to the research targets.

Answer:

We have added (reference [1]) the information  relevant to the ecological frame in the Chechen republic and the importance of garlic and onion for the Chechen population diet.

Materials and Methods

- Table 1 should be removed, and the temperature and relative humidity data should be included inside the text. 

Answer:

We have deleted Table 1 and reported temperature and relative humidity values inside the text, as recommended above.

Results and Discussion

- In Tables 5 and 8 the line sequence should be the same as in the other Tables: Control, AMF, Se, AMF+Se.

- Please, check if the combination between AMF and Se has been expressed uniformly as AMF+Se inside the text as well as the Tables and Figures captions.

Answer:

We have performed corrections and check according to the above comments.

References

- Some other references could be added and cited.  

Colella, T.; Candido, V.; Campanelli, G.; Camele, I.; Battaglia, D. Effect of irrigation regimes and artificial mycorrhization on insect pest infestations and yield in tomato crop. Phytoparasitica 2014, 42, 235-246. DOI: 10.1007/s12600-013-0356-3.            It could be cited in “Introduction” section” (line  59, along with citation n. 4).

22 Castronuovo D., Russo D., Libonati R., Faraone I., Candido V., Picuno P., Andrade P., Valentao P., Milella L., 2019. Influence of shading treatment on yield, morphological traits and phenolic profile of sweet basil (Ocimum basilicum L.). Scientia Horticulturae, 254, 91-98, doi: 10.1016/j.scienta.2019.04.077               It could be cited in “Material and Methods” section” (line  153, along with citation n. 20).

Answer:

We have added the references recommended above inside the text and the References section.

Dear Reviewer 2, 

Thank you very much for your useful recommendations aimed to improve our manuscript. We addressed all your comments and performed the relevant amendments and/or modifications inside the text and tables, highlighting them in red color.

Answers to Reviewer 2 comments

The manuscript needs minor changes and corrections as outlined below.

Abstract

Concise information about the experimental protocol should be reported.

The conclusion sentence should be completed with further short details.

Answer:

The experimental details have been added and the conclusion has been expanded, according to the above recommendations.

Keywords: put in full and in italics style “ sativum” and “A. cepa” (Allium sativum, Allium cepa”.

Answer:

 We have performed the corrections in compliance with the above comments.

3) Introduction

- Some information about the importance of Allium cepa L. and Allium sativum L. in the area where the research was carried out should be reported.

- A reference to the investigation area should be added to the research targets.

Answer:

We have added (reference [1]) the information  relevant to the ecological frame in the Chechen republic and the importance of garlic and onion for the Chechen population diet.

Materials and Methods

- Table 1 should be removed, and the temperature and relative humidity data should be included inside the text. 

Answer:

We have deleted Table 1 and reported temperature and relative humidity values inside the text, as recommended above.

Results and Discussion

- In Tables 5 and 8 the line sequence should be the same as in the other Tables: Control, AMF, Se, AMF+Se.

- Please, check if the combination between AMF and Se has been expressed uniformly as AMF+Se inside the text as well as the Tables and Figures captions.

Answer:

We have performed corrections and check according to the above comments.

References

- Some other references could be added and cited.  

Colella, T.; Candido, V.; Campanelli, G.; Camele, I.; Battaglia, D. Effect of irrigation regimes and artificial mycorrhization on insect pest infestations and yield in tomato crop. Phytoparasitica 2014, 42, 235-246. DOI: 10.1007/s12600-013-0356-3.            It could be cited in “Introduction” section” (line  59, along with citation n. 4).

22 Castronuovo D., Russo D., Libonati R., Faraone I., Candido V., Picuno P., Andrade P., Valentao P., Milella L., 2019. Influence of shading treatment on yield, morphological traits and phenolic profile of sweet basil (Ocimum basilicum L.). Scientia Horticulturae, 254, 91-98, doi: 10.1016/j.scienta.2019.04.077               It could be cited in “Material and Methods” section” (line  153, along with citation n. 20).

Answer:

We have added the references recommended above inside the text and the References section.

e other references could be added and cited.  

Colella, T.; Candido, V.; Campanelli, G.; Camele, I.; Battaglia, D. Effect of irrigation regimes and artificial mycorrhization on insect pest infestations and yield in tomato crop. Phytoparasitica 2014, 42, 235-246. DOI: 10.1007/s12600-013-0356-3.            It could be cited in “Introduction” section” (line  59, along with citation n. 4).

22 Castronuovo D., Russo D., Libonati R., Faraone I., Candido V., Picuno P., Andrade P., Valentao P., Milella L., 2019. Influence of shading treatment on yield, morphological traits and phenolic profile of sweet basil (Ocimum basilicum L.). Scientia Horticulturae, 254, 91-98, doi: 10.1016/j.scienta.2019.04.077               It could be cited in “Material and Methods” section” (line  153, along with citation n. 20).

Answer:

We have added the references recommended above inside the text and the References section.